## RESEARCH ARTICLE

# Differences in three-dimensional kinetic determinants of jump height between single- and double-leg countermovement jumps

**Terumitsu Miyazaki[1,\*], Shota Yamamoto[1,2] and Hirotomo Kubota[1]**

## ABSTRACT

Single-leg jumping tasks are directly linked to scoring in various sports; therefore, identifying the biomechanical determinants of higher single-leg jumping performance is crucial for athletic performance. Single-leg jumping can utilize the hip and lumbosacral joints in three dimensions. However, the joint kinetic variables in the frontal and horizontal planes associated with jump height are not fully understood. This study aimed to investigate the differences in joint kinetic factors associated with jump height between single- and double-leg countermovement jumps (CMJs) using waveform statistical analysis. Forty-eight male collegiate athletes performed single- and double-leg CMJs, and lower-limb and lumbosacral joint kinetics were analyzed. The results showed that the extension torque and flexion–extension power of the lower-limb joints were significantly correlated with jump height in single- and double-leg CMJs. Moreover, during the propulsive phase of the single-leg CMJ, greater hip abduction torque, lumbosacral lateral flexion torque, and hip and lumbosacral axial rotation torques were positively correlated with jump height, but not in the double-leg CMJ. The results highlight different mechanisms for achieving higher jump height between single- and double-leg CMJs. The findings suggest that strength training and movement modification targeting the frontal- and horizontal-plane motions may improve single-leg CMJ performance.

KEY WORDS: Inverse dynamics, Joint power, Joint torque, Lower limb, Waveform

## INTRODUCTION

Single-leg jumping tasks are frequently observed in a variety of sports activities, such as basketball, volleyball, soccer, and track and field. Single-leg jumping performance is directly linked to scoring and competitive outcomes (e.g. lay-up shots in basketball and jump distance in the long jump). Thus, clarifying the biomechanical determinants of single-leg jumping performance may contribute to improving athletic performance.

Among the various jumping modes, athletes commonly perform countermovement jumps (CMJs). The jump height of CMJs is widely used to assess athletic performance (Kale et al., 2009; Nagahara et al., 2014; Wing et al., 2020). Previous studies have investigated the relationships between CMJ jump height and biomechanical factors, including joint kinetics (Aragón-Vargas and

Gross, 1997; Cleather et al., 2013; Shinchi et al., 2024) and muscle strength (Chang et al., 2015; Kozinc and Šarabon, 2022; Tsiokanos et al., 2002). These studies indicate that extension torque and flexion–extension power exerted by the lower-limb joints are important determinants of jump height in both single- and double-leg CMJs (Aragón-Vargas and Gross, 1997; Cleather et al., 2013; Shinchi et al., 2024).

The current understanding of joint kinetics during single- and double-leg CMJs is limited to sagittal-plane motions. In the single-leg CMJ, the contralateral leg is unconstrained, which allows greater three-dimensional motion of the hip joint, lumbosacral joint, and pelvis. Accordingly, joint kinetics in horizontal and frontal planes may be associated with jump height in the single-leg CMJ. The hip joint exerts larger abduction torques and the lumbosacral joint exerts lateral flexion torque during the single-leg squat jump compared with the double-leg squat jump (Sado et al., 2020). Additionally, torques and powers have been observed across multiple axes in the single-leg running jump (Sado et al., 2018), high jump (Fujimori et al., 2024), long jump (Funken et al., 2019), and sprinting (Sado et al., 2019; Schache et al., 2011). These findings suggest that frontal- and horizontal-plane motions would be essential for moving the body forward and upward during single-leg tasks. Therefore, the torques and powers generated by the hip and lumbosacral joints in the abduction–adduction and axial rotational axes are hypothesized to be associated with jump height of single-leg CMJ.

Furthermore, the free-leg side elevation of the pelvis contributes to the generation of mechanical energy related to jump height in single-leg squat jump, whereas it does not in double-leg squat jump (Sado et al., 2020). From a mechanical perspective, the free-leg side elevation of the pelvis can be caused by the hip abduction torque of the stance leg and lumbosacral lateral flexion torque. These torques may contribute to jump height through the pelvic elevation. A similar mechanical characteristic may also be observed during single-leg CMJ, but not double-leg CMJ, suggesting that the kinetic factors associated with jump height may differ between single- and double-leg CMJs. Understanding the differences in kinetic determinants of jump height could contribute to the development of specific training programs for single-leg CMJ; however, the differences remain unclear.

To clarify the biomechanical differences between single- and double-leg CMJs, it is important to identify the specific timing at which joint kinetic variables are associated with jump height. In sports biomechanics research, waveform statistical analysis using one-dimensional statistical parametric mapping (Pataky, 2010) has been used to identify the specific timing at which kinetic variables are associated with athletic performance during sprinting (Colyer et al., 2018; Nagahara and Murata, 2024). This approach can provide a better understanding of joint function at specific time points. Although previous studies have examined relationships between jump height and discrete kinetic variables (Aragón-Vargas and Gross, 1997; Cleather et al., 2013; Shinchi et al., 2024), the specific timing at which joint kinetic variables are associated with the jump height in

[1]National Institute of Fitness and Sports in Kanoya, Kanoya, 891-2393 Kagoshima, Japan. [2]Biwako Seikei Sport College, Otsu, 520-0503 Shiga, Japan.

*Author for correspondence (t-miyazaki@nifs-k.ac.jp)

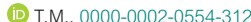 T.M., 0000-0002-0554-3122

CMJs remains unclear. Barker et al. (2018) demonstrated that vertical ground reaction force (GRF) variables during the propulsive phase of CMJ are major determinants of jump height. Thus, the joint torques and powers during the propulsive phase may also be associated with the jump height of CMJs.

Therefore, the present study aimed to examine the differences in joint kinetic variables associated with jump height between single- and double-leg CMJs. It was hypothesized that the hip abduction torque and lumbosacral lateral flexion torque, as well as their powers, during the propulsive phase of single-leg CMJ would be associated with jump height, but not double-leg CMJ.

## RESULTS
### Jump height and vertical GRF
The mean jump heights were 0.183±0.038 m (range, 0.114–0.295 m) and 0.379±0.056 m (range, 0.258–0.488 m) for the single- and double-leg CMJs, respectively (Fig. 3A). The vertical GRF during the propulsive phase was greater in the double-leg CMJ than in the single-leg CMJ (Fig. 3B).

### Kinematics of lower-limb joints, lumbosacral joint, and pelvis
The flexion–extension angles of the lower-limb joints and the lumbosacral joint exhibited similar waveform patterns between the single- and double-leg CMJs (Fig. 4A,B,C,F). The hip joint angles in the abduction–adduction and axial rotation axes and the lumbosacral joint angles in the lateral flexion and axial rotation axes exhibited different waveform patterns (Fig. 4D,E,G,H). In the single-leg CMJ, the hip joint adducted and then abducted from the unweighting phase to the propulsive phase (Fig. 4D). Additionally, the lumbosacral joint laterally flexed from the stance-leg side toward the free-leg side during the propulsive phase in the single-leg CMJ (Fig. 4G).

Regarding pelvic kinematics, the anterior–posterior angle and angular velocity exhibited similar waveform patterns between the single- and double-leg CMJs (Fig. 5A,D). In the frontal plane, pelvic elevation on the free-leg side was observed, whereas this motion was not observed in the double-leg CMJ (Fig. 5B,E).

### Lower-limb joint kinetics with SPM-1D results
Similar waveform patterns in the extension torque and flexion–extension power of the ankle, knee, and hip joints were observed

between the single- and double-leg CMJs (Fig. 6), whereas the hip torques and powers in the abduction–adduction and axial rotation axes differed (Fig. 6I,J,N,O). Hip abduction torque was observed throughout the take-off phase of the single-leg CMJ (Fig. 6I). Positive hip abduction–adduction power was also observed, and these torques and powers tended to be greater than those in the double-leg CMJ (Fig. 6N).

The statistical results of SPM-1D are shown in Fig. 6, and the SPM *t*-curves are available in Figs S1 and S2. Ankle plantarflexion and knee extension torques exhibited positive correlations with jump height in the single- and double-leg CMJs during the propulsive phase (ankle: 83%–96% and 84%–97%; knee: 81%–90% and 87%–95%, for the single- and double-leg CMJs, respectively; Fig. 6F,G). Similar correlations were also observed for ankle and knee joint powers in the single- and double-leg CMJs (ankle: 92%–99% and 90%–99%; knee: 90%–92% and 92%–96%, for the single- and double-leg CMJs, respectively; Fig. 6K,L).

Regarding hip joint kinetics, extension torque exhibited positive correlations with jump height in the double-leg CMJ during the propulsive phase (69%–92%; Fig. 6H). No significant correlations were observed in the single-leg CMJ during this phase. Hip joint torques in the abduction–adduction and axial rotation axes exhibited positive correlations with jump height in the single-leg CMJ during the propulsive phase (abduction–adduction: 80%–88%; internal–external rotation: 63%–79%; Fig. 6I,J). Similar correlations were also observed for joint powers in the abduction–adduction axis (87%–95%; Fig. 6N). No significant correlations were observed for these torques and powers in the double-leg CMJ (Fig. 6I,J,N,O).

Furthermore, significant correlations were observed between jump height and the kinetic variables of some lower-limb joints during the unweighting phase of the single- and double-leg CMJs (e.g. knee extension torque in the double-leg CMJ: 0%–9%; hip extension torque in the single-leg CMJ: 6%–23%; Fig. 6).

### Lumbosacral joint kinetics with SPM-1D results
Similar waveform patterns in the extension torque and flexion–extension power of the lumbosacral joint were observed between the single- and double-leg CMJs (Fig. 7A,D,G), whereas the lateral flexion and axial rotation torques and powers differed (Fig. 7E,F,H,I). During the propulsive phase of the single-leg CMJ, lumbosacral lateral flexion torque and power were observed and tended to be greater than those in the double-leg CMJ (Fig. 7E,H).

The statistical results of SPM-1D are shown in Fig. 7, and the SPM *t*-curves are available in Figs S3 and S4. Torque and power in the flexion–extension axis exhibited positive correlations with jump height in the double-leg CMJ during the propulsive phase (69%–91%; Fig. 7D). No significant correlations during the propulsive phase were observed for the kinetic variables in the other axes in both CMJs.

### Associations between jump height and discrete kinetic variables
The peak torques and powers in the flexion–extension axis of the ankle, knee, hip, and lumbosacral joints were positively correlated with jump height in the single- and double-leg CMJs (Table 1). Peak hip abduction torque was positively correlated with jump height in both CMJs, whereas peak hip abduction–adduction power was positively correlated with jump height only in the single-leg CMJ (Table 1). Additionally, the peak torques and powers of hip axial rotation and lumbosacral lateral flexion were significantly correlated with jump height only in the single-leg CMJ (Table 1).

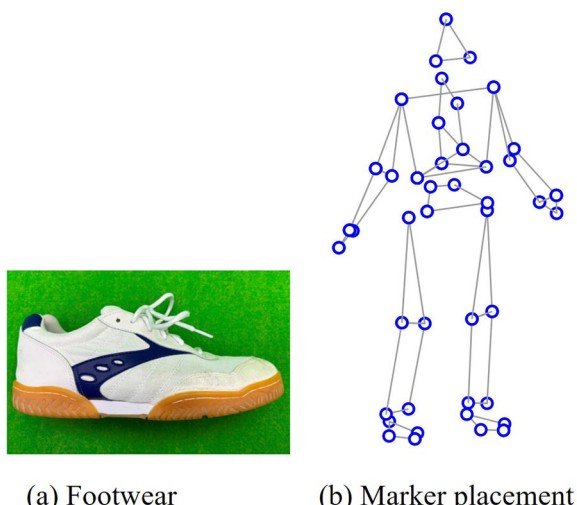

(a) Footwear          (b) Marker placement

**Fig. 1. Standardized footwear (A) and marker placement (B).**

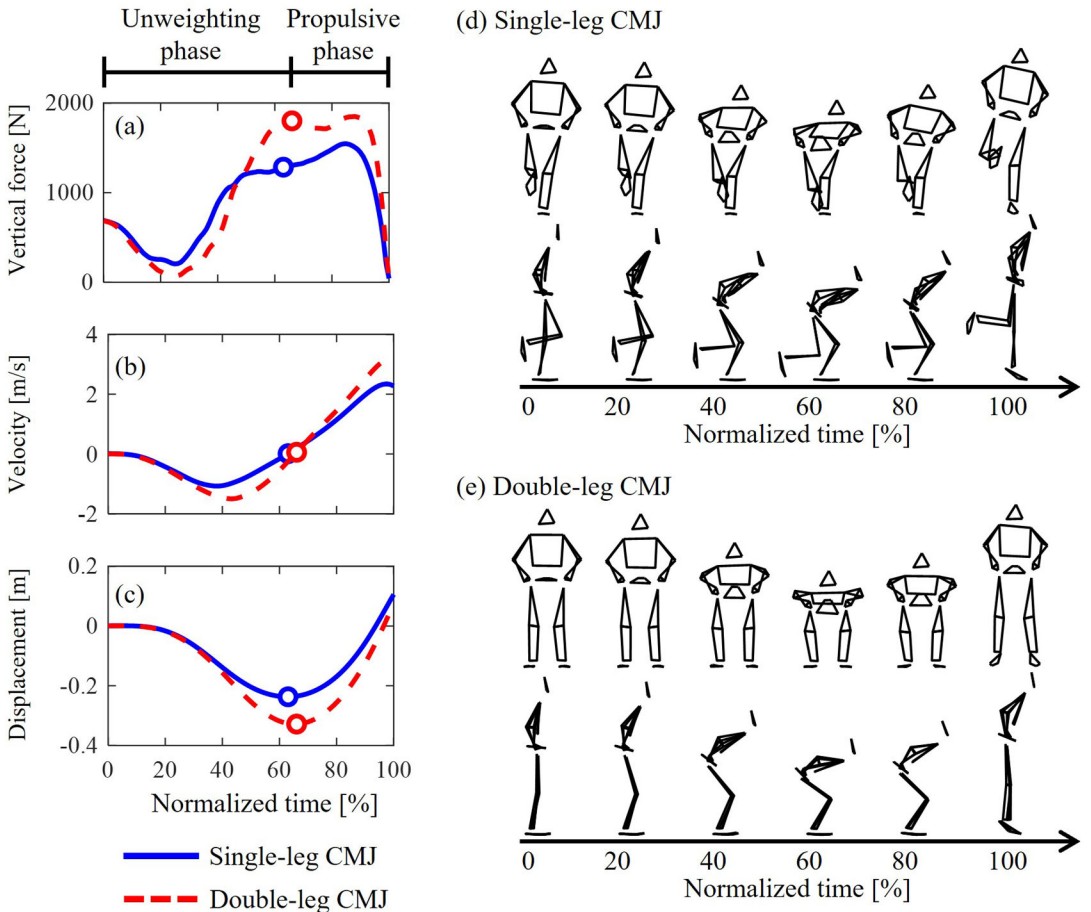

**Fig. 2. Definition of the unweighting and propulsive phases of single- and double-leg CMJs.** Typical data and stick images are presented: vertical component of GRF (A), vertical CoM velocity, vertical CoM displacement (C), and stick images in sagittal and frontal planes (D,E). The circle markers (A–C) indicate when the CoM velocity was zero, representing the transition from the unweighting to the propulsive phase.

A multiple regression analysis using peak torques (Table 2) showed that the predictors of jump height differed between the single- and double-leg CMJs. In the double-leg CMJ, the predictors were peak ankle plantarflexion and hip extension torques, whereas in the single-leg CMJ, the predictors were peak ankle plantarflexion, hip abduction, lumbosacral extension, and lateral flexion torques (Table 2). Additionally, the same predictors were selected in the analysis using peak powers in both CMJs (Table 3).

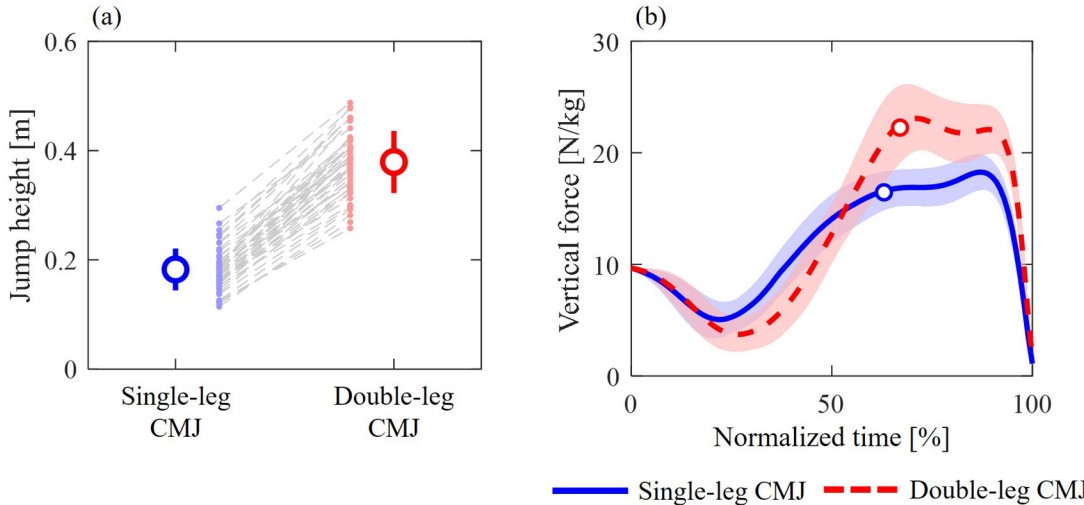

**Fig. 3. Mean and individual values of jump heights (A) and vertical component of GRF (B) during single-leg (blue solid line) and double-leg (red dashed line) CMJs.** The circle markers indicate when the CoM velocity was zero, representing the transition from the unweighting to the propulsive phase.

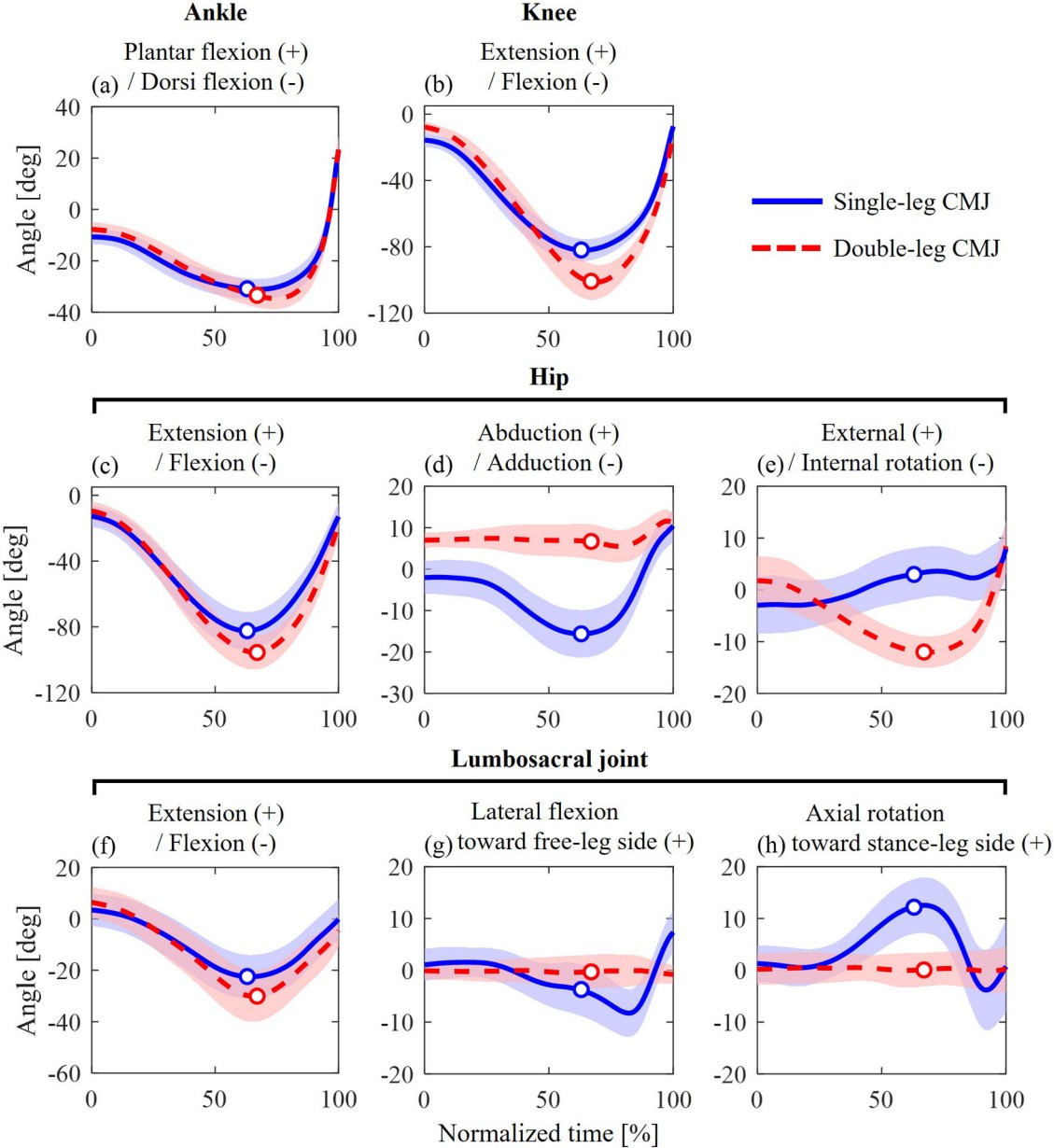

**Fig. 4. Joint angles of the lower-limb (A–E) and lumbosacral joints (F–H) during single-leg (blue solid line) and double-leg (red dashed line) CMJs.** The circle markers indicate when the CoM velocity was zero, representing the transition from the unweighting to the propulsive phase.

## DISCUSSIONS

To the best of our knowledge, this is the first study that investigates the differences in lower-limb and lumbosacral joint kinetic determinants associated with jump height between single- and double-leg CMJs using waveform statistical analysis. The present study found that, during the propulsive phase of the single-leg CMJ, greater hip abduction torque, lumbosacral lateral flexion torque, and hip and lumbosacral axial rotation torques were associated with higher jump height, whereas these associations were not observed in the double-leg CMJ. Similar results were observed for the hip and lumbosacral joint powers. These results support our hypothesis. Moreover, the waveform profiles of joint kinematics and kinetics showed similar patterns in the sagittal plane but different patterns in the frontal and horizontal planes between single- and double-leg CMJs. These profiles imply that coordinating three-dimensional motions of the hip-pelvis-trunk complex may be required for better performance in the single-leg CMJ. Therefore, the findings suggest that joint kinetic variables in the frontal and horizontal planes are associated with single-leg CMJ performance, in addition to those in the sagittal plane, implying that the kinetic determinants of achieving higher jump height differ between single- and double-leg CMJs.

This study found that greater hip abduction torque and abduction–adduction power during the propulsive phase were associated with higher jump height in the single-leg CMJ, but not in the double-leg CMJ. Greater peak torque and power of lumbosacral lateral flexion were associated with higher jump height in the single-leg CMJ, whereas this association was not observed in the double-leg CMJ. These results suggest that, for the single-leg CMJ, the kinetic variables of the hip and lumbosacral joints in the frontal and horizontal planes were associated with higher jump height, in addition to extension torques and flexion–extension powers in the lower-limb joints. The hip abduction and lumbosacral lateral flexion

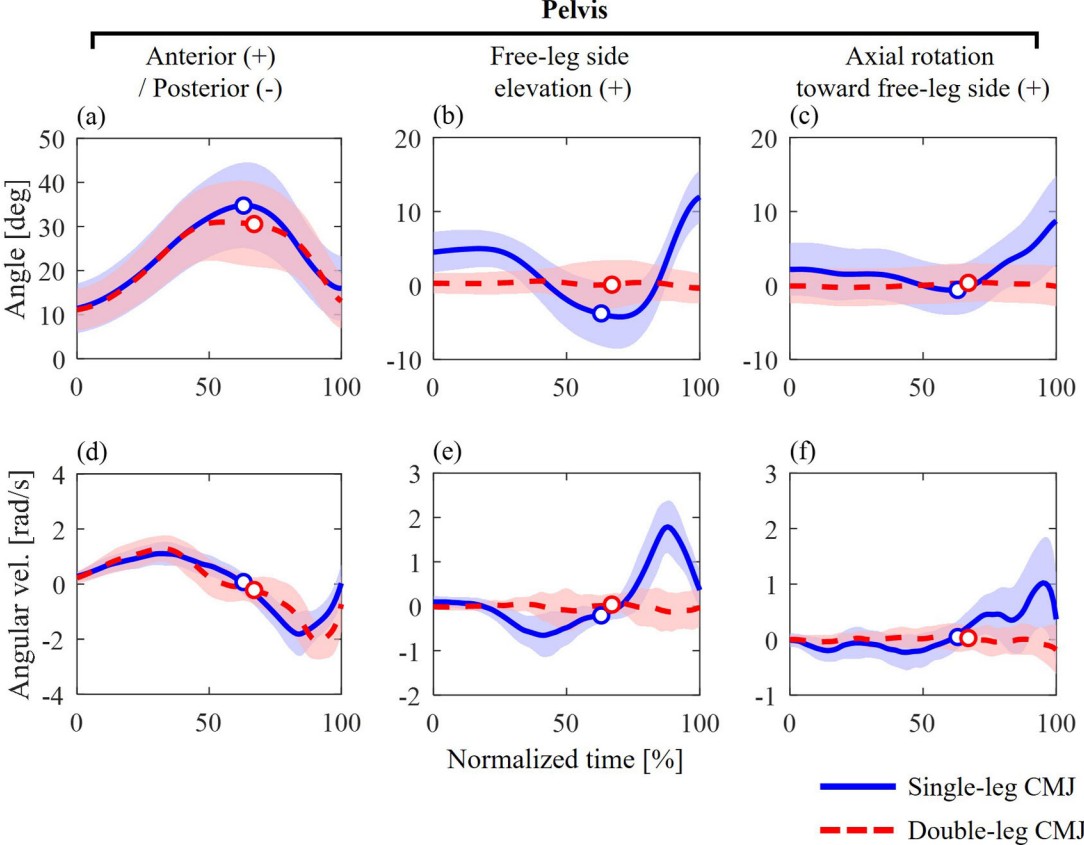

**Fig. 5. Segment angles (A–C) and angular velocities (D–F) of the pelvis during single-leg (blue solid line) and double-leg (red dashed line) CMJs.** The circle markers indicate when the CoM velocity was zero, representing the transition from the unweighting to the propulsive phase.

torques induce pelvic rotation, mainly the free-leg side elevation. The peak angular velocity of pelvic elevation was observed during the propulsive phase, similar to the timing of the peak torques and powers of the hip and lumbosacral joints. The pelvic elevation, which generates whole-body mechanical energy, contributes more to jump height in a single-leg squat jump than in a double-leg squat jump (Sado et al., 2020). Thus, in the single-leg CMJ, the hip abduction and lumbosacral lateral flexion torques may be associated with higher jump height by inducing pelvic rotation in the frontal plane.

Furthermore, in the single-leg CMJ, the correlation coefficients between peak hip joint torques and jump height were larger for abduction ($r$=0.640) than for extension ($r$=0.462). The same pattern was observed for joint powers at these axes (hip abduction–adduction power, $r$=0.576; hip flexion–extension power, $r$=0.472). Moreover, the multiple regression analysis using peak joint powers selected flexion–extension powers as the predictors in the single- and double-leg CMJs. In contrast, the multiple regression analysis using peak torques showed that the jump height of the single-leg CMJ was predicted by ankle plantarflexion, hip abduction, lumbosacral extension, and lumbosacral lateral flexion torques, but the hip and knee extension torques were not selected. These predictors differed from those for the double-leg CMJ as follows: jump height was predicted by ankle plantarflexion and hip extension torques. These results suggest that the hip and knee extension torques have a weaker association with jump height in the single-leg CMJ than in the double-leg CMJ. Taken together with our findings and a previous study (Shinchi et al., 2024), hip extension torque may be more strongly associated with jump height than ankle and knee kinetic variables in the double-leg CMJ; however, in the single-leg CMJ, our

findings suggest that hip abduction and lumbosacral lateral flexion torques were more strongly associated with jump height than hip extension torque. Consistent with this finding, a previous study reported that the correlation coefficient between the isometric hip extension torque and the jump height of the single-leg CMJ was $r$=0.321, whereas that for the double-leg CMJ was $r$=0.437 (Kozinc and Šarabon, 2022). Thus, improving the muscle strength for hip abductors and lumbosacral lateral flexors might be associated with improved single-leg CMJ performance, in addition to strengthening lower-limb extensors.

In the horizontal plane, the present study found that the axial rotation torques and powers at the hip and lumbosacral joints were associated with higher jump height in the single-leg CMJ, whereas these associations were not observed in the double-leg CMJ. Because the rotation torques mainly induce pelvic axial rotation, these torques might be associated with jump height by enhancing swinging motion of the free leg. A previous study showed that the jump height decreased by 2.7 cm for the right leg and 2.4 cm for the left leg under conditions with restriction of free-leg swing compared to conditions without restriction (Schmidt et al., 2024), indicating that free-leg swinging affects the jump height. Free-leg swing is a fundamental skill for the single-leg CMJ, and differences in task demands may explain the discrepant results between single- and double-leg CMJs. Additionally, pelvic axial rotation has a smaller contribution to the jump height than pelvic elevation in the single-leg CMJ (Sado et al., 2020). Therefore, greater hip internal rotation and lumbosacral axial rotation torques could enhance free-leg swing by increasing pelvic axial rotation, suggesting that these torques might be indirectly associated with jump height in the single-leg CMJ.

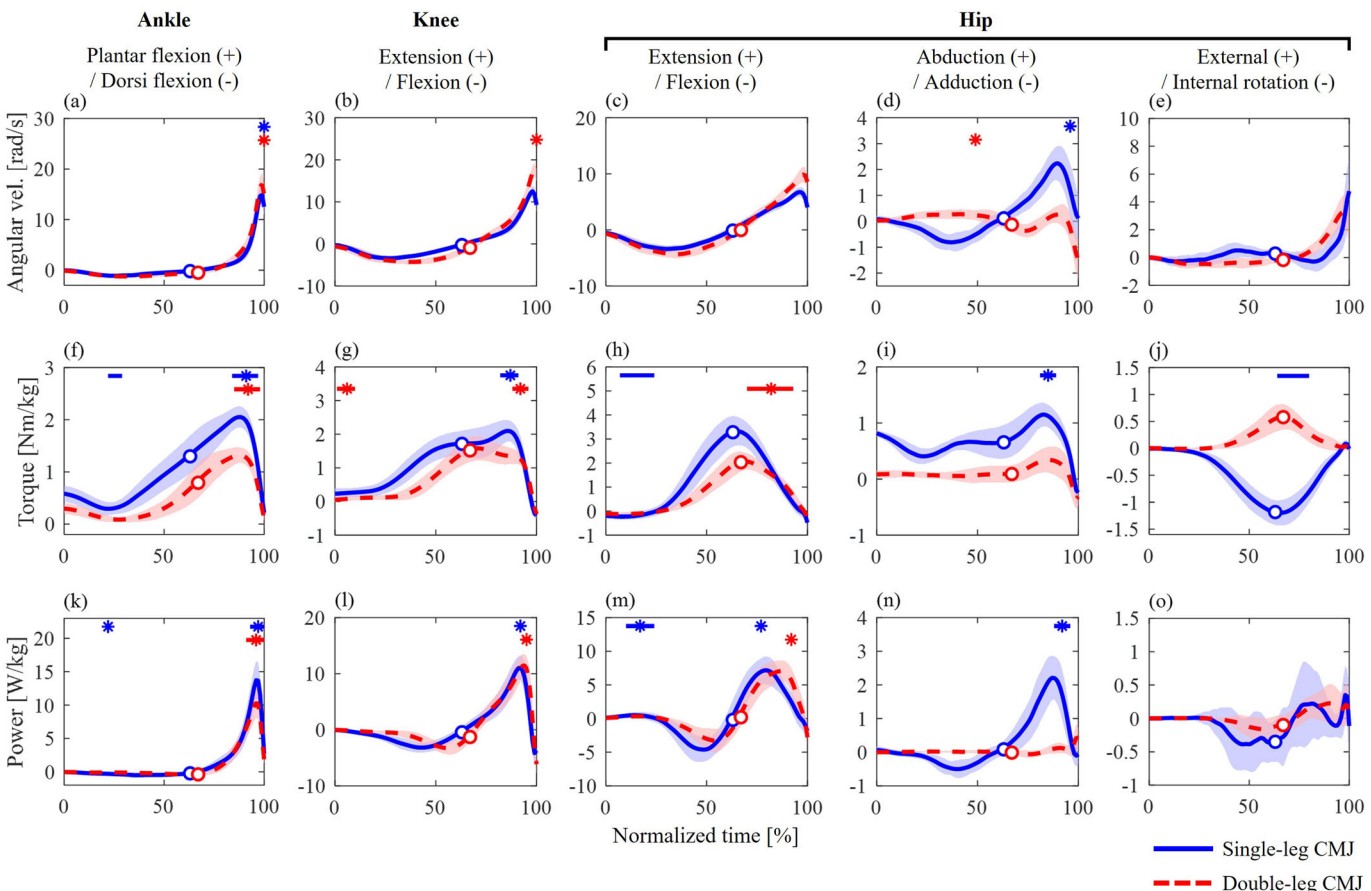

**Fig. 6. Joint angular velocities, torques, and powers of the ankle (A,F,K), knee (B,G,L), and hip (C–E,H–J,M–O) joints during single-leg (blue solid line) and double-leg (red dashed line) CMJs.** The circle markers indicate when the CoM velocity was zero, representing the transition from the unweighting to the propulsive phase. The asterisks (*) and horizontal bars at the top of each graph indicate the time points showing significant correlations with jump height. The asterisks and bars represent positive correlations, whereas the bars alone represent negative correlations. The blue and red lines represent single- and double-leg CMJs, respectively.

In addition to the above-mentioned results during the propulsive phase, this study found significant correlations between jump height and the kinetics of specific lower-limb joints during the unweighting phase of both single- and double-leg CMJs. The results suggest that these joint torques and powers may influence jump height by regulating the depth of the center of mass (CoM). An additional analysis (Fig. 8) showed that vertical CoM displacement was 0.094±0.035 m smaller in the single-leg CMJ than in the double-leg CMJ. This difference may affect trunk and lower-limb postures during the unweighting phase and at the onset of the propulsive phase (Figs 4 and 8). Thus, differences in joint kinetics between single- and double-leg CMJs might be attributed to differences in countermovement characteristics. Additionally, because CoM depth was not standardized between participants in this study, CoM displacement showed large inter-individual differences in both single- and double-leg CMJs: the ranges of maximum depth were −0.377 to −0.166 and −0.471 to −0.212, respectively. These inter-individual differences may be associated with inter-individual variability in joint kinetics during both unweighting and propulsive phases. Although inter-individual relationships between the jump height and unweighting-phase kinematics remain unclear, several studies (Mandic et al., 2015; Pérez-Castilla et al., 2021) have shown that changes in the depth of the CoM during this phase affect GRF variables and jump height. Taken together, lower-limb joint torques and powers during the unweighting phase may influence jump height

by regulating CoM kinematics. Nevertheless, the effects of joint kinetics on jump height during this phase remain unclear. Further research is warranted to clarify these mechanisms.

The single-leg CMJ is a substantially different task from the double-leg CMJ. In particular, the magnitude of countermovement, swing-leg involvement, and postural stability demands differ between single- and double-leg CMJs. Although greater hip abduction and lumbosacral lateral flexion torques were associated with higher jump height in the single-leg CMJ, these torques may be linked to higher jump height through their role in regulating free-leg swing and maintaining postural stability. Furthermore, although the present study analyzed the leg with the higher jump height (the superior leg) in the single-leg CMJ to represent each participant's best performance, an additional analysis of the contralateral leg (the inferior leg) showed both similarities and differences in the correlation results between the superior and inferior legs (Table S1, Figs S5–S8). The inferior leg did not show a significant time interval in which hip abduction torque and abduction–adduction power were correlated with jump height. In contrast, the peak values of hip abduction torque, hip adduction–abduction power, and lumbosacral lateral flexion torque were positively associated with the inferior-leg jump height, consistent with those of the superior leg. Therefore, these findings indicate that greater hip abduction and lumbosacral lateral flexion torques and their powers were associated with higher jump height of the single-leg CMJ, even in the inferior leg. However, inter-limb differences in

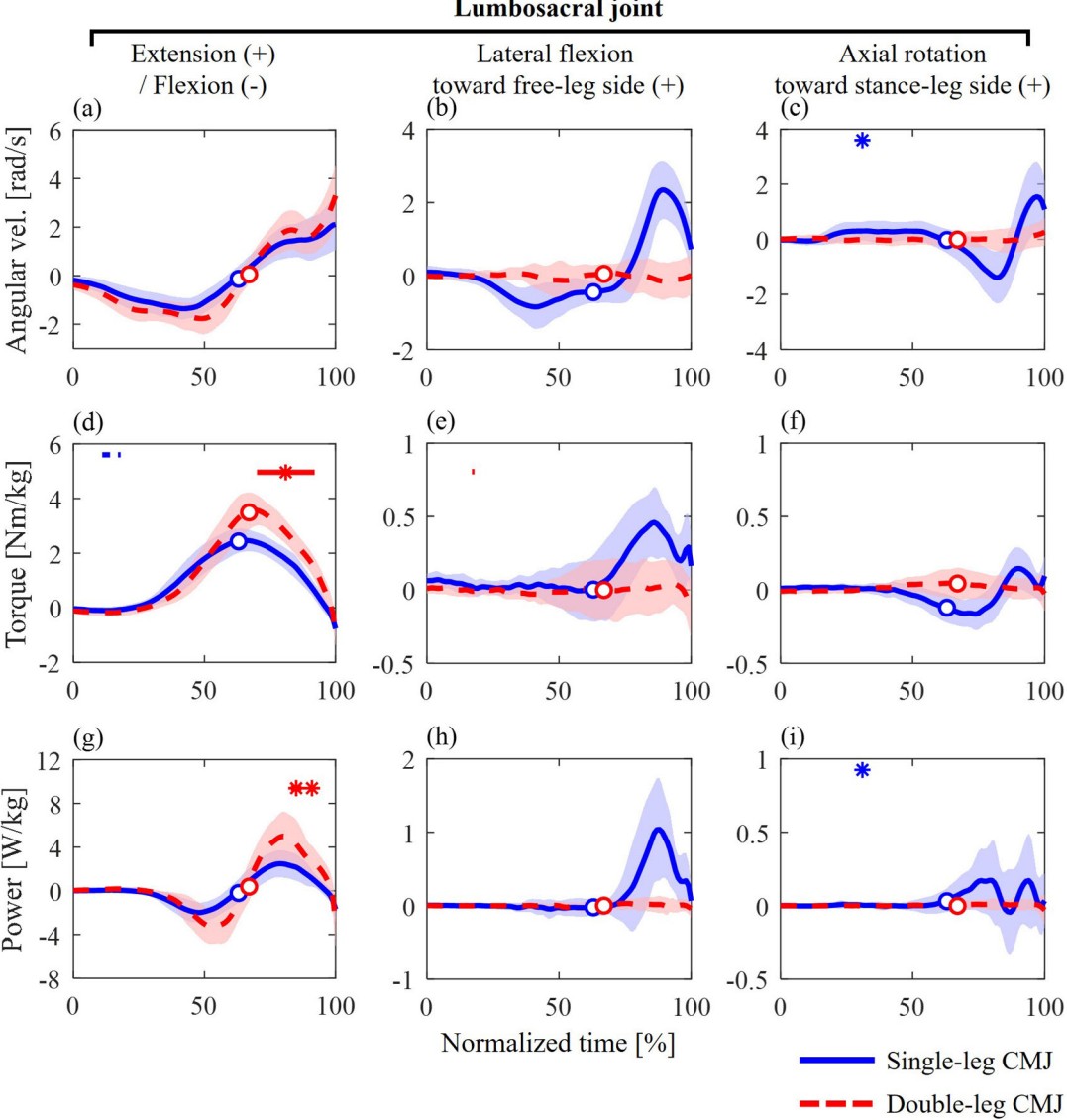

**Fig. 7. Joint angular velocities, torques, and powers of the lumbosacral joint during single-leg (blue solid line) and double-leg (red dashed line) CMJs.** The circle markers indicate when the CoM velocity was zero, representing the transition from the unweighting to the propulsive phase. The asterisks (*) and horizontal bars at the top of each graph indicate the time points showing significant correlations with the jump height. The asterisks and bars represent positive correlations, whereas the bars alone represent negative correlations. The blue and red lines represent single- and double-leg CMJs, respectively.

countermovement characteristics or postural stability may influence the kinetic determinants of jump height; thus, inter-limb differences in jumping mechanics should be examined in future studies.

As a practical implication of the present study, strength training for the hip abductors and lumbosacral lateral flexors, along with training for the lower-limb extensors, may help improve the performance of single-leg CMJ. These muscles contribute to pelvic elevation through the coordinated generation of hip abduction and lumbosacral lateral flexion torques. Consistently, previous studies have shown the importance of these torques in other single-leg jumping tasks, including running, high jump, and long jump (Fujimori et al., 2024; Funken et al., 2019; Sado et al., 2018). Furthermore, Nagahara et al. (2025) found that greater pelvic list strength could be an indicator of sprint performance. Therefore, in addition to strength training, enhancing the neuromuscular coordination in the hip and lumbosacral joints may be associated with higher single-leg CMJ performance,

and may also have implications for performance in other single-leg motor tasks.

This study has several limitations. First, because this study was cross-sectional, it remains unclear whether improvements in frontal- and horizontal-plane motions of the hip and lumbosacral joints are directly linked to improvements in single-leg CMJ performance. Thus, longitudinal intervention studies would be warranted to address this limitation. Second, although this study recruited male athletes from various sports backgrounds to enhance generalizability, sex and sports background may influence our findings. Previous studies have reported that GRF variables and joint kinematics differ between female and male athletes during jumping tasks (Cronström et al., 2016). Such sex-related differences may lead to differences in the joint kinetic determinants of CMJ jump height. Additionally, we recruited participants from several sports backgrounds. In particular, three marine-sport athletes (3/48 participants), whose sports did not

**Table 1. Correlation coefficients (r) and *P*-values between the jump height and peak joint torque and power of lower-limb and lumbosacral joints during single- and double-leg CMJs**

| | | Single-leg CMJ | | Double-leg CMJ | |
|---|---|---|---|---|---|
| | | r | *P*-value | r | *P*-value |
| Peak torque (unit, Nm/kg) | Ankle plantarflexion (+) | **0.731** | **<0.001** | **0.586** | **<0.001** |
| | Knee extension (+) | **0.628** | **<0.001** | **0.390** | **0.006** |
| | Hip extension (+) | **0.462** | **0.001** | **0.473** | **0.001** |
| | Hip abduction (+) | **0.640** | **<0.001** | **0.418** | **0.003** |
| | Hip external rotation (+) | – | – | 0.227 | 0.120 |
| | Hip internal rotation (−) | **−0.440** | **0.002** | – | – |
| | Lumbosacral extension (+) | **0.409** | **0.004** | **0.486** | **<0.001** |
| | Lumbosacral lateral flexion toward the free-side leg (+) | **0.317** | **0.028** | 0.096 | 0.514 |
| | Lumbosacral axial rotation toward the stance-leg side (+) | **0.371** | **0.009** | 0.250 | 0.087 |
| Peak positive power (unit, W/kg) | Ankle plantar–dorsiflexion axis | **0.769** | **<0.001** | **0.723** | **<0.001** |
| | Knee flexion–extension axis | **0.677** | **<0.001** | **0.587** | **<0.001** |
| | Hip flexion–extension axis | **0.472** | **0.001** | **0.419** | **0.003** |
| | Hip abduction–adduction axis | **0.576** | **<0.001** | 0.062 | 0.677 |
| | Hip external–internal rotation axis | **0.565** | **<0.001** | 0.231 | 0.115 |
| | Lumbosacral flexion–extension axis | **0.404** | **0.004** | **0.440** | **0.002** |
| | Lumbosacral lateral flexion axis | **0.391** | **0.006** | 0.256 | 0.079 |
| | Lumbosacral axial rotation axis | 0.141 | 0.338 | 0.093 | 0.531 |

The bold font indicates a statistically significant correlation ($P<0.05$). The correlation results of the hip external rotation torque for single-leg CMJ and hip internal rotation torque for double-leg CMJ are not shown because these torques were not produced (see Fig. 6).

involve jumping tasks, were included; however, an additional analysis excluding these three athletes ($n=45$) showed similar associations between jump height and joint kinetics to those observed in the full sample ($n=48$). Because single-leg CMJs require three-dimensional motion of the hip and lumbosacral joints, the fundamental movement pattern may be similar across sex and sports backgrounds. However, the generalizability of our findings across these factors remains unclear. Finally, performance level may also affect our findings. Shinchi et al. (2024) suggested that elite athletes can generate superior hip extension torque during the double-leg CMJ. The jump height (0.379±0.056 m) and peak hip extension torque (2.2±0.35 Nm/kg) in the present study were lower than those reported in a previous study (0.437±0.063 m and 2.4±0.5 Nm/kg, respectively; Shinchi et al., 2024). Thus, the sports background and performance level should be considered when interpreting our findings.

## MATERIALS AND METHODS
### Participants
Forty-eight male collegiate athletes (age, 19.8±1.4 years; height, 1.72±0.05 m; body mass, 68.0±8.3 kg) participated in this study. The participants participated in various college sports clubs, including soccer, baseball, track and field, volleyball, and marine sports, and were habitually engaged in sports activities. All participants reported no pain during maximal effort athletic activities, including jumping and sprinting. Furthermore, all participants had no musculoskeletal injuries in their trunk or lower limbs. The study protocol and its purpose were explained to all participants, and written informed consent was obtained from each participant. This study

was approved by the Ethics Committee of the National Institute of Fitness and Sports in Kanoya (reference number: 22-1-51).

### Experiments and data collection
The participants wore standardized clothing and footwear (Fig. 1A; Lucky Bell Co., Ltd., Japan). This footwear consisted of standard athletic gym shoes that did not have a rocker sole, excessive cushioning, and a carbon plate as recently seen in 'super shoes' (Healey et al., 2022; Hébert-Losier & Pamment, 2023). The 48 retroreflective markers were attached to the trunk and limbs (Fig. 1B; Miyazaki and Fujii, 2025a,b). The warm-up protocol included 5 min of cycling on an ergometer (POWER MAX, Konami Sports Co., Ltd., Japan; 2.0 kp at 60 rpm), dynamic stretching of the lower-limb joints, and several practice trials of the jumping task at submaximal and maximal effort levels. This warm-up protocol was designed by the authors and piloted prior to the experiment. The warm-up familiarized participants with the CMJ task and helped participants perform maximal-effort CMJs throughout experiments. After warm-up, the participants performed three types of jumping tasks at maximal effort: double-leg CMJ and single-leg CMJs on the right and left sides. Each task was performed until three successful trials were obtained, up to a maximum of five trials. The participants were instructed to perform CMJ with a self-selected countermovement depth while keeping their arms alongside the trunk to minimize the influence of arm swing. They were not instructed to restrict swing-leg motion. Additionally, they maintained a static standing posture for 2–3 s to minimize body movement before initiating CMJs. This procedure was implemented to accurately calculate the velocity data from the GRFs. An adequate recovery time between trials and sessions was provided to avoid the effects of fatigue on jumping performance.

**Table 2. Results of stepwise multiple regression analyses using peak torques of lower-limb joints and lumbosacral joint to predict jump height in the single- and double-leg CMJs**

| Independent variables | $\beta$ | Equation | Adjusted $R^2$ |
|---|---|---|---|
| Single-leg CMJ | | | |
| $x_1$: ankle plantarflexion | 0.510 ($P<0.001$) | $y=0.100x_1+0.046x_2+0.021x_3+0.038x_4-0.169$ | 0.669 ($P<0.001$) |
| $x_2$: hip abduction | 0.267 ($P=0.017$) | | |
| $x_3$: lumbosacral extension | 0.233 ($P=0.011$) | | |
| $x_4$: lumbosacral lateral flexion | 0.186 ($P=0.038$) | | |
| Double-leg CMJ | | | |
| $x_1$: ankle plantarflexion | 0.498 ($P<0.001$) | $y=0.187x_1+0.056x_2-0.005$ | 0.432 ($P<0.001$) |
| $x_2$: hip extension | 0.347 ($P=0.004$) | | |

$\beta$, standardized partial regression coefficient; $y$, jump height.

**Table 3. Results of stepwise multiple regression analyses using peak powers of lower-limb joints and lumbosacral joint to predict jump height in the single- and double-leg CMJs**

| Independent variables | $\beta$ | Equation | Adjusted $R^2$ |
|---|---|---|---|
| Single-leg CMJ | | $y=0.007x_1+0.005x_2+0.004x_3+0.006x_4-0.028$ | 0.779 ($P<0.001$) |
| $x_1$: ankle plantar-dorsi flexion | 0.546 ($P<0.001$) | | |
| $x_2$: knee flexion-extension | 0.270 ($P=0.003$) | | |
| $x_3$: hip flexion-extension | 0.247 ($P=0.002$) | | |
| $x_4$: lumbosacral flexion-extension | 0.195 ($P=0.009$) | | |
| Double-leg CMJ | | $y=0.012x_1+0.010x_2+0.009x_3+0.006x_4-0.038$ | 0.792 ($P<0.001$) |
| $x_1$: ankle plantar-dorsi flexion | 0.497 ($P<0.001$) | | |
| $x_2$: lumbosacral flexion-extension | 0.383 ($P<0.001$) | | |
| $x_3$: knee flexion-extension | 0.355 ($P<0.001$) | | |
| $x_4$: hip flexion-extension | 0.175 ($P=0.018$) | | |

$\beta$, standardized partial regression coefficient; $y$, jump height.

The marker trajectories during the jumping tasks were recorded at 250 Hz using a 16-camera three-dimensional motion capture system (Mac3D, Motion Analysis Corp., USA). GRF data were simultaneously collected at 1000 Hz using two force plates (9287AB, Kistler Inc., Switzerland). These systems were synchronized.

### Data analysis
Data analysis was performed using MATLAB software (MATLAB 2022b, MathWorks Inc., USA). Marker trajectories were low-pass filtered using a fourth-order zero-lag Butterworth digital filter with a cutoff frequency of 15 Hz. GRF data were also low-pass filtered using the same cutoff frequency (Kristianslund et al., 2012). These filtered data were used to calculate lower-limb and lumbosacral joint kinematics and kinetics. The kinematic and kinetic data were extracted from the initiation of the jump to take-off. The initiation of the jump was defined as the instance when the vertical component of the GRF was less than 1.0% of body weight (Shinchi et al., 2024; Yamashita et al., 2020). Take-off was defined as the instance when the vertical GRF was less than 20 N (Krzyszkowski et al., 2022).

For single-leg CMJ, the leg (right or left) with the higher mean jump height across the three trials was selected as the representative side for analysis (right leg: 26 participants; left leg: 22 participants). For double-leg CMJ, the kinematic and kinetic variables of the lower-limb joints, including the joint angle, angular velocity, joint torque, and joint power, were averaged between the right and left legs.

The jump height of CMJs was calculated from the vertical GRF (Fig. 2A). First, the vertical acceleration of the CoM was calculated according to Eqn (1). Second, the vertical CoM velocity (Fig. 2B) was calculated by time-integrating the vertical acceleration. Third, jump height was computed from the vertical CoM velocity at take-off using Eqn (2). In addition, vertical CoM displacement (Fig. 2C) was calculated by time-integrating the vertical

CoM velocity:

$$Acceleration\ [m/s^2] = (F_z - mg)/m, \qquad (1)$$

$$Jump\ height\ [m] = v_{toff}^2/2g, \qquad (2)$$

where $F_z$ is the vertical GRF, $m$ is body mass, $g$ is gravitational acceleration (9.8 m/s²), and $v_{toff}$ is vertical CoM velocity at take-off. The vertical GRF was not low-pass filtered prior to the calculations of jump height, in accordance with previous studies (Shinchi et al., 2024; Yamashita et al., 2020). Two phases were defined using the vertical CoM velocity: an unweighting phase and a propulsive phase (Fig. 2). The unweighting phase was defined as the period from the initiation of the jump to the instance when the vertical CoM velocity reached zero. At this instant (vertical CoM velocity=0 m/s), the CoM reaches its lowest vertical position approximately 60% of the jumping phase (Fig. 2). The propulsive phase was defined as the period from this instance to take-off.

The whole body was modelled as a rigid-linked skeletal model (Winter, 2009) in line with our previous studies (Miyazaki and Fujii, 2025a,b), incorporating 15 body segments: head–neck, upper trunk, pelvis, upper arm, forearm, hand, thigh, shank, and foot. These segments were connected at 14 joints in total: neck, lumbosacral, shoulder, elbow, wrist, hip, knee, and ankle joints. The elbow, wrist, ankle, and knee joint centers were defined as midpoints between the lateral and medial markers attached to the segments. The shoulder and neck joint centers were defined in accordance with the method of Reed et al. (1999). The lumbosacral and hip joint centers were defined in accordance with the method of Sado et al. (2024). The body segment inertia parameters were defined based on previous studies (Dumas et al., 2007b,a). The inertia parameters were normalized to each participant's body mass and segment length (Dumas et al., 2007b,a).

The procedures for calculating kinematic and kinetic variables were based on the prior literature describing standard biomechanics methods (Winter, 2009).

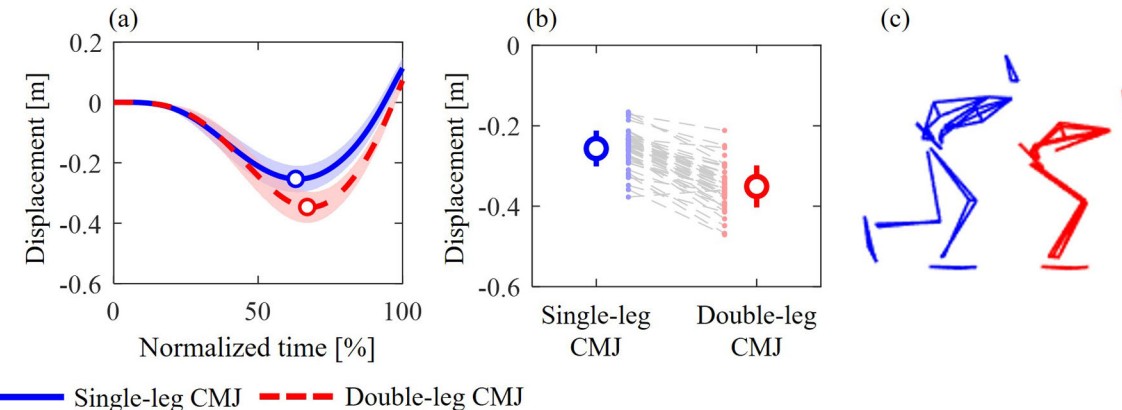

**Fig. 8. The vertical CoM displacement (A), the maximum depth of the CoM displacement (B), and typical stick images at the maximum depth (C) during single-leg (blue) and double-leg (red) CMJs.** The circle markers indicate when the CoM velocity was zero, representing the transition from the unweighting to the propulsive phase.

The torques of the lumbosacral, hip, knee, and ankle joints were calculated using inverse dynamics problem-solving (Winter, 2009). The powers of these joints were calculated as the dot product of joint torque and joint angular velocity (Winter, 2009). These joint kinetic variables were normalized by the participant's body mass. The pelvis segment, lumbosacral joint, and lower-limb joint angles were calculated as Cardan angles. The variables were time-normalized to 101 time points from the initiation of the jump to take-off. The time-normalized data of the joint angular velocities, torques, and powers of the lower-limb joints (hip, knee, and ankle joints) and the lumbosacral joint were extracted for subsequent statistical waveform analysis.

## Statistical analysis

The outcome variables were defined as the mean of three trials for each CMJ task, providing a representative value for each participant. Descriptive data were presented as means±s.d. Statistical significance was set at 0.05 for all analyses.

To examine the relationships between jump height and joint kinetic variables (joint angular velocity, torque, and powers of the hip, knee, ankle, and lumbosacral joints) in single- and double-leg CMJs, correlation analysis using SPM-1D (Pataky, 2010) was performed by utilizing the spm-1d software package (version M.0.4.9, https://spm1d.org/) in MATLAB.

Pearson product correlation coefficients between jump height and conventional discrete variables (e.g. peak torque and power of hip, knee, ankle, and lumbosacral joints) were calculated in single- and double-leg CMJs. Moreover, a stepwise multiple regression analysis was performed for each CMJ task to examine whether the predictive models for jump height differed between single- and double-leg CMJs. In this analysis, the jump height was set as the dependent variable, and the peak torques and powers of lower-limb joints and lumbosacral joint were set as independent variables. To minimize the effects of multicollinearity among predictors, especially between torque and power, the stepwise multiple regression analyses were performed separately using peak torques and peak powers. The correlation and stepwise multiple regression analyses were performed using the Statistics and Machine Learning Toolbox in MATLAB 2020b.

## Conclusion

This study identified differences in the kinetic factors of lower-limb and lumbosacral joints associated with the jump height between single- and double-leg CMJs. Although the extension torque and flexion–extension power generated by the lower-limb joints were associated with the jump height in both CMJs, greater hip abduction and lumbosacral lateral flexion torques were associated with higher jump height during the propulsive phase of the single-leg CMJ. Additionally, the axial rotation torques generated by the hip and lumbosacral joints were associated with the jump height in single-leg CMJ. These findings indicate that frontal- and horizontal-plane motions of the hip and lumbosacral joints, in addition to sagittal-plane motion, might be associated with jump height in the single-leg CMJ.

## Acknowledgements
We would like to thank the members at the National Institute of Fitness and Sports in Kanoya for their help in collecting the data.

## Competing interests
The authors declare no competing or financial interests.

## Author contributions
Conceptualization: T.M., H.K.; Data curation: T.M., S.Y.; Formal analysis: T.M., S.Y., H.K.; Funding acquisition: T.M.; Investigation: T.M., S.Y.; Methodology: T.M., S.Y., H.K.; Project administration: T.M.; Resources: T.M.; Software: T.M.; Supervision: T.M.; Validation: T.M., H.K.; Visualization: T.M.; Writing – original draft: T.M.; Writing – review & editing: T.M., S.Y., H.K.

## Funding
This work was supported by the Japan Society for the Promotion of Science (JSPS) KAKENHI grant number [24K20565]. Open Access funding provided by National Institute of Fitness and Sports in Kanoya. Deposited in PMC for immediate release.

## Data and resource availability
Data will be made available on request. All relevant data and details of resources can be found within the article and its supplementary information.

## Ethical considerations
This study was approved by the Ethics Committee of the National Institute of Fitness and Sports in Kanoya (reference number: 22-1-51). A written informed consent was obtained from all participants prior to the experiment.

## Peer review history
The peer review history is available online at https://journals.biologists.com/bio/lookup/doi/10.1242/bio.062434.reviewer-comments.pdf

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
