## [Peer Review File · Biology Open]

Differences in three-dimensional kinetic determinants of jump height between single- and double-leg countermovement jumps

Shota Yamamoto, Hiroto Kubota and Terumitsu Miyazaki
10.1242/bio.062434

Editor: Lewis Halsey

Review timeline

Original submission:	18 December 2025
Editorial decision:	6 January 2026
First revision received:	2 March 2026
Accepted:	9 April 2026

Original submission

First decision letter

MS ID#: bio.062434

MS Title: Differences in three-dimensional kinetic determinants of jump height between single- and double-leg countermovement jumps

Authors: Terumitsu Miyazaki; Shota Yamamoto; Hiroto Kubota

I have now reached a decision on the above manuscript.

The reviewer reports are shown at the bottom of this email.

As you will see, the reviewers raised a number of substantial criticisms that prevent me from accepting the paper at this stage.

They suggest, however, that a revised version might prove acceptable, if you can address their concerns. If you think that you can deal satisfactorily with the criticisms on revision, I would be pleased to see a revised manuscript. We would then return it to the reviewers.

At this stage, we also ask you to ensure your manuscript complies with our formatting guidelines. Provided you are able to fully address the referees' comments, we are positive about publication of your paper (we accept over 95% of revision submissions) and therefore hope you won't mind any extra work involved in reformatting your manuscript at this point.

Please upload both a 'clean' version of your Word file, along with a highlighted version clearly showing where you have made changes in the revised manuscript. Please avoid using 'Track changes' in Word files as these are lost in PDF conversion.

I should be grateful if you would also provide a point-by-point response detailing how you have dealt with the points raised by the reviewers in the 'Response to Reviewers' box. Please attend to all of the reviewers' comments. If you do not agree with any of their criticisms or suggestions please explain clearly why this is so.

Reviewer 1

Comments for the author

The authors present a thorough comparison between the kinematics and kinetics of single and double leg countermovement jumps, finding significant differences in how maximum jump heights are achieved through control of the lower limb joints. This has interesting implications for sport biomechanics and training protocols. While the paper is well-written and somewhat well-reasoned, I feel that the lack of control in the experimental design means the conclusions of the study may be over-exaggerated. I think this could be improved by recognising these limitations further and perhaps caveating the study conclusions.

Introduction

- Why are single-leg countermovement jumps interesting? The opening paragraphs mention CMJs generally but mostly in contexts which would require a double-leg jump. Sports which may require a single-leg jump are mentioned in the Discussion, but also warrant mentioning in the Introduction.

Methods

- The participants jumped with a self-selected countermovement depth. Did you measure or quantify this? Did this vary much between the participants? Jump performance is heavily influenced by the degree of countermovement performed, so the difference in jump height between single and double leg jump would be partially related to the inability to perform as much of a countermovement during a single leg jump (I would assume), due to issues with balance. It would be interesting to see some reports of the amount of countermovement performed between the two jumps, and how they correlate with jump height (i.e. can single leg jump be improved by increasing countermovement), or the degree of lumbosacral rotation/abduction.

- More information is needed on how jump height was calculated from vGRF and velocity, to help readers who are not experts in jumping mechanics. Given you have built a model of each participant with mass and inertial properties, I assume that body CoM could also have been calculated and used to quantify height?

- In the model, were body segment inertial parameters changed for each individual?

- The "kinetic" variables included in the SPM analysis need to be specified. Were they reduced to one variable for this analysis?

Results

- Normalised jump heights by the degree of countermovement performed could also give indications on how much these heights were influenced by this.

Discussion

- Did you restrict free-leg swing in your study? It doesn't seem like you did, but it's not totally clear. Is "swing" referring to movement in all hip joint degrees of freedom? Sagittal plane swinging would help jump height through the mass and inertia of the swinging limb, while movement in other degrees of freedom would help from a balance perspective. If it were allowed to move, could the kinematics of the swinging limb be quantified here?

- Could the increased lateral flexion and axial rotation of the lumbosacral joint be due to the need to balance during the single-leg jump, rather than anything particularly functionally important? There seems to be a lot of variation in the angles, velocities, powers etc. suggesting that some participants were better at balancing here than others.

Figures

- Figure 1- This would be much more informative if it showed still images of various phases of the two types of jump, so the reader can get a better understanding of what was done. This could be photographs of the participants or images of the motion capture data at these time points. You could even highlight the unweighted and propulsive phases here. Showing an image of the shoes is not particularly interesting, and one image of the motion capture markers with no context is also not very informative.

- Figure 2- Graphs showing the amount of countermovement performed in each jump and jump heights would be informative here, alongside vGRFs.

Reviewer 2

Comments for the author

General comments:

This study addresses an interesting and relevant question regarding the biomechanics of single- and double-leg countermovement jumps (CMJs), with potential implications for understanding athletic performance. However, the manuscript would benefit from careful revision of the English throughout, particularly with respect to grammar, spelling and sentence clarity. While the scientific content is largely clear, there are numerous minor language issues that reduce readability and occasionally obscure meaning, I strongly recommend a thorough language revision prior to resubmission.

Abstract:

* The opening sentences would benefit from more clearly stating why this study is valuable. At present, the motivation for the work is not sufficiently established. Consider briefly emphasising the importance of SMJ performance in sport and why understanding its biomechanical determinants is meaningful.

* While the main findings are described, the broader implications are not clearly articulated. It would strengthen the abstract to explicitly state why these results matter for biomechanics research or applied training practice.

Introduction:

The introduction is generally well structured and addresses three key points:

1. Increasing understanding of the biomechanics of single- and double-leg CMJs is valuable. The importance and relevance of CMJs are clearly described in the first paragraph.
2. There are biomechanical differences between single- and double-leg CMJs. This is addressed in the second and third paragraphs, although this key message only becomes apparent several lines in. I recommend revising the opening sentences of these paragraphs to make this distinction clearer from the outset.
3. There is limited understanding regarding the timing of kinetic variables associated with jump height and identifying these may be important. This knowledge gap is outlined in paragraph four.

However, clarity could be improved in several places. Several sentences would benefit from rephrasing for clarity, particularly where technical concepts are introduced, and there are a number of minor grammatical issues (e.g. subject-verb agreement, repeated words) that should be addressed throughout the introduction. Specific points include:

* Line 68: clarify the phrasing (e.g. "countermovement jumps are..." or "jumping is...").

* Lines 70-71: consider explicitly stating that improved biomechanical understanding may inform coaching and technique development, which may in turn enhance athletic performance.

- * Line 75: this may be clearer if rephrased as "However, current understanding of ... is limited to...".
- * Lines 76-78: I found the explanation here to be a little unclear; please clarify.
- * Lines 79-80: please clarify what is meant by "important determinants" (e.g. determinants of jump height?).
- * Line 80: "are" should be used instead of "is".
- * Lines 87-88: consider revising to "...and powers have been observed...".
- * Lines 94-95: the word "contribute" is used twice; revising this sentence may improve clarity and flow.

Methods:

- * The participant cohort includes athletes who routinely perform jumping movements (e.g. volleyball players) and those who do not (e.g. undefined 'marine sports'). Differences in training background could plausibly influence technique and performance. Have analyses been conducted excluding participants who do not regularly train jumping movements and if so, are the results consistent? This issue is briefly acknowledged later in the manuscript, but I feel that it warrants stronger consideration here.
- * No female participants were recruited. Given the well-documented sex bias in biomechanics and sports science research, this is a notable limitation. Please provide a justification for this decision and explicitly acknowledge it as a limitation.
- * The standardized footwear is shown in Figure 1a, but its characteristics are not described. Please clarify whether this was a specific model and provide relevant details (e.g. cushioning, sole stiffness).
- * Several methodological procedures would benefit from clearer referencing to established protocols:
 - o Warm-up procedure: does this follow a standard protocol? If so, please cite a reference.
 - o Line 165: please provide a reference to support the GRF threshold of <20 N.
 - o Lines 178-179: "The whole body was modeled as a rigid-linked skeletal model..." Please provide a reference for this modelling approach. Also note the incorrect spelling of "modelled".
 - o Lines 188-192: does this analysis follow an established protocol? If so, please cite it.
- * The countermovement depth was self-selected. Please discuss whether this lack of standardization may have influenced inter-individual variability and the observed correlations.
- * For single-leg CMJs, the leg with the higher mean jump height was selected for analysis. Please justify this choice and clarify whether left-right comparisons were performed or considered.

Statistical analysis:

Correlation analyses may be appropriate for the stated aims, but please consider whether additional or complementary analyses (e.g. accounting for multicollinearity or shared variance between joints) could provide further insight.

Results:

- * The Results section is quite dense and may be difficult for readers to follow. Consider restructuring this section using clearer subheadings and/or shorter paragraphs to guide the reader.

* Figure 2 contains a spelling error ("Sigle-leg CMJ").

* Please ensure that all figures and tables are clearly referenced at appropriate points in the main text.

Discussion:

* The interpretation of frontal- and horizontal-plane kinetics in single-leg CMJs is thoughtful and well linked to prior literature. However, some statements regarding applicability to sports performance (e.g. running, high jump, long jump) should be more cautiously framed, given that not all participants were trained jump specialists. This limitation is acknowledged briefly, but I recommend expanding this discussion to more clearly delimit the populations to which the findings can be generalized.

* The Discussion occasionally reiterates the same findings across multiple paragraphs. Condensing these sections and focusing more explicitly on interpretation and implications would improve clarity and flow.

* While the authors note the cross-sectional design, several parts of the Discussion describe correlational relationships in mechanistic or causal terms. I recommend more consistently framing these findings as associations and clarifying that the observed kinetic patterns may reflect successful strategies rather than direct drivers of performance.

* Statements extending the findings to improvements in other single-leg motor tasks (e.g. running or sport-specific performance) should be more clearly framed as speculative, as these outcomes were not directly assessed in the present study.

In summary, this is a promising and technically strong study that addresses an important gap in the literature. Addressing the points above would substantially improve clarity, methodological transparency and interpretability.

Reviewer's Responses to Questions

Experimental quality

Does each figure have the proper controls?

If 'No', please indicate reasons in Comments for Author box below.

Reviewer #1:

- Yes

Reviewer #2:

- Yes

Were the data analyzed using appropriate statistical tests?

If 'No', please indicate reasons in Comments for Author box below.

Reviewer #1:

- Yes

Reviewer #2:

- Yes

Reproducibility

Were experiments performed using adequate number of biological replicates?

If 'No', please indicate reasons in Comments for Author box below.

Reviewer #1:

- Yes

Reviewer #2:

- Yes

Does the methods section provide sufficient detail to permit reproducibility?

If 'No', please indicate reasons in Comments for Author box below.

Reviewer #1:

- No

Reviewer #2:

- Yes

Completeness

Are the manuscript's conclusions supported by the data?

If 'No', please indicate reasons in Comments for Author box below.

Reviewer #1:

- Yes

Reviewer #2:

- Yes

Scholarship

Do the authors cite and discuss the merits of data that would argue for and against their conclusion?

If 'No', please indicate reasons in Comments for Author box below.

Reviewer #1:

- Yes

Reviewer #2:

- Yes

Does the manuscript title & abstract accurately reflect the contents of the manuscript, without hyperbole?

If 'No', please indicate reasons in Comments for Author box below.

Reviewer #1:

- Yes

Reviewer #2:

- Yes

First revision

Author response to reviewers' comments

Manuscript ID: bio.062434

**Differences in three-dimensional kinetic determinants of jump height
between single- and double-leg countermovement jumps
Terumitsu Miyazaki, Shota Yamamoto, and Hiroto Kubota**

We thank the reviewers for providing useful comments on our paper. We sincerely believe that the comments have helped us improve the quality of our paper. We have revised the manuscript in accordance with these comments. In the revised manuscript, corresponding changes to the manuscript are highlighted in yellow.

Reviewer #1

The authors present a thorough comparison between the kinematics and kinetics of single and double leg countermovement jumps, finding significant differences in how maximum jump heights are achieved through control of the lower limb joints. This has interesting implications for sport biomechanics and training protocols. While the paper is well-written and somewhat well-reasoned, I feel that the lack of control in the experimental design means the conclusions of the study may be over-exaggerated. I think this could be improved by recognising these limitations further and perhaps caveating the study conclusions.

Response: Thank you very much for your valuable suggestions. We have addressed your suggestions with point-by-point responses and revised the manuscript accordingly.

Introduction

- Why are single-leg countermovement jumps interesting? The opening paragraphs mention CMJs generally but mostly in contexts which would require a double-leg jump. Sports which may require a single-leg jump are mentioned in the Discussion, but also warrant mentioning in the Introduction.

Response: We appreciate this comment. We have revised the opening paragraph of the Introduction to explain why single-leg jumping in sports activities is important.

Revised text (Page 4, Lines 67-71):

“Single-leg jumping tasks are frequently observed in a variety of sports activities, such as basketball, volleyball, soccer, and track and field. Single-leg jumping performance is directly linked to scoring and competitive outcomes (e.g., lay-up shots in basketball and jump distance in the long jump). Thus, clarifying the biomechanical determinants of single-leg jumping performance may contribute to improving athletic performance.”

Methods

- The participants jumped with a self-selected countermovement depth. Did you measure or quantify this? Did this vary much between the participants? Jump performance is heavily influenced by the degree of countermovement performed, so the difference in jump height between single and double leg jump would be partially related to the inability to perform as much of a countermovement during a single leg jump (I would assume), due to issues with balance. It would be interesting to see some reports of the amount of countermovement performed between the two jumps, and how they correlate with jump height (i.e. can single leg jump be improved by increasing countermovement), or the degree of lumbosacral rotation/abduction.

Response: Thank you for this valuable comment. Countermovement characteristics may influence jump performance in both single- and double-leg CMJ, although their relationship with jump height remains unclear. We agree that differences in countermovement characteristics between single- and double-leg CMJs may partly contribute to the differences in correlation results observed in this study. Therefore, we have added an additional analysis of countermovement characteristics (Figure 8) and expanded the Discussion accordingly.

Revised text (Page 13, Lines 379-388):

“An additional analysis (Figure 8) showed that vertical CoM displacement was 0.094 ± 0.035 m smaller in the single-leg CMJ than in the double-leg CMJ. This difference may affect trunk and lower-limb postures during the unweighting phase and at the onset of the propulsive phase (Figures 4 and 8). Thus, differences in joint kinetics between single- and double-leg CMJs might be attributed to differences in countermovement characteristics. Additionally, because CoM depth was not standardized between participants in this study, CoM displacement showed large inter-individual differences in both single- and double-leg CMJs: the ranges of maximum depth were -0.377 to -0.166 and -0.471 to -0.212, respectively. These inter-individual differences may be associated with inter-individual variability in joint kinetics during both unweighting and propulsive phases.”

- More information is needed on how jump height was calculated from vGRF and velocity, to help readers who are not experts in jumping mechanics. Given you have built a model of each participant with mass and inertial properties, I assume that body CoM could also have been calculated and used to quantify height?

Response: We appreciate this comment. We have added a more detailed description of the calculation method for jump height. Specifically, jump height was calculated from the vertical velocity of the center of mass at take-off, which was computed from the vertical GRF. This approach has been widely used in previous studies.

Revised text (Page 7 Lines 175-184):

“The jump height of CMJs was calculated from the vertical GRF (Figure 2a). First, the vertical acceleration of the center of mass (CoM) was calculated according to Eq. (1). Second, the vertical CoM velocity (Figure 2b) was calculated by time-integrating the vertical acceleration. Third, jump height was computed from the vertical CoM velocity at take-off using Eq. (2). In addition, vertical CoM displacement (Figure 2c) was calculated by time-integrating the vertical CoM velocity.

$$\text{Acceleration [m/s}^2\text{]} = (F_z + mg)/m \quad \text{Eq. (1)}$$

$$\text{Jump height [m]} = v_{\text{toff}}^2/2g \quad \text{Eq. (2)}$$

where F_z is the vertical GRF, m is body mass, g is gravitational acceleration (9.8 m/s^2), and v_{toff} is vertical CoM velocity at take-off. The vertical GRF was not low-pass filtered prior to the calculations of jump height, in accordance with previous studies (Shinchi et al., 2024; Yamashita et al., 2020).”

- In the model, were body segment inertial parameters changed for each individual?

Response: We appreciate this comment. We have revised the Methods section to clarify that the body segment inertial parameters were individualized for each participant.

Revised text (Page 8, Lines 199-200):

“The inertia parameters were normalized to each participant’s body mass and segment length (Dumas et al., 2007b, 2007a).”

- The “kinetic” variables included in the SPM analysis need to be specified. Were they reduced to one variable for this analysis?

Response: We thank the reviewer for this comment. We have added detailed information in the method section.

Revised text (Page 8, Lines 208-210):

“The time-normalized data of the joint angular velocities, torques, and powers of the lower-limb joints (hip, knee, and ankle joints) and the lumbosacral joint were extracted for subsequent statistical waveform analysis.”

Revised text (Page 8, Lines 216-217):

“... (joint angular velocity, torque, and powers of the hip, knee, ankle, and lumbosacral joints) ...”

Results

- Normalised jump heights by the degree of countermovement performed could also give indications on how much these heights were influenced by this.

Response: We appreciate this valuable suggestion. We agree that normalizing jump height by the degree of countermovement might provide an additional perspective on the influence of countermovement characteristics on CMJ performance. To address this comment, we added further discussion on the differences in countermovement between single- and double-leg CMJs and included an additional analysis (**Figure 8; Page 13, Lines 379-388**). However, we did not add the suggested normalized jump height, because this variable is difficult to interpret in this study. As our primary aim was to examine the associations between jump height and joint kinetics, we consider this additional analysis to be valuable for future research.

Discussion

- Did you restrict free-leg swing in your study? It doesn't seem like you did, but it's not totally clear. Is "swing" referring to movement in all hip joint degrees of freedom? Sagittal plane swinging would help jump height through the mass and inertia of the swinging limb, while movement in other degrees of freedom would help from a balance perspective. If it were allowed to move, could the kinematics of the swinging limb be quantified here?

Response: We thank the reviewer for this comment. First, we have added information in the Methods section to clarify that free-leg motion was not restricted during the single-leg CMJ (**Page 6, Lines 150-151**). Second, although free-leg motion was allowed during the single-leg CMJ, we did not analyze the kinematics of the swinging limb in this study. As our main aim was to examine the associations between jump height and the joint kinetics of the stance leg and lumbosacral joint, this additional analysis was beyond the scope of the current manuscript. However, we agree that analysis of the swinging limb would be useful and should be examined in future research. To address this comment, we have added further discussion of this point in the Discussion (**Page 13, Lines 395-400**).

Revised text (Page 6, Lines 150-151):

“They were not instructed to restrict swing-leg motion.”

Revised text (Page 13, Lines 395-400):

“The single-leg CMJ is a substantially different task from the double-leg CMJ. In particular, the magnitude of countermovement, swing-leg involvement, and postural stability demands differ between single- and double-leg CMJs. Although greater hip abduction and lumbosacral lateral flexion torques were associated with higher jump height in the single-leg CMJ, these torques may be linked to higher jump height through their role in regulating free-leg swing and maintaining postural stability.”

- Could the increased lateral flexion and axial rotation of the lumbosacral joint be due to the need to balance during the single-leg jump, rather than anything particularly functionally important? There seems to be a lot of variation in the angles, velocities, powers etc. suggesting that some participants were better at balancing here than others.

Response: Thank you for this valuable suggestion. Single-leg CMJ may require greater postural stability than double-leg CMJ. We agree that the increased lateral flexion and axial rotation of the lumbosacral joint may be related to balance control, and that inter-individual differences in postural stability may affect single-leg CMJ performance. To address this point, we added further discussion in the Discussion section.

Revised text (Pages 13-14, Lines 395-413):

“The single-leg CMJ is a substantially different task from the double-leg CMJ. In particular, the magnitude of countermovement, swing-leg involvement, and postural stability demands differ between single- and double-leg CMJs. Although greater hip abduction and lumbosacral lateral flexion torques were associated with higher jump height in the single-leg CMJ, these torques may be linked to higher jump height through their role in regulating free-leg swing and maintaining postural stability. ... However, inter-limb differences in countermovement characteristics or postural stability may influence the kinetic determinants of jump height; thus, inter-limb differences in jumping mechanics should be examined in future studies.”

Figures

- Figure 1- This would be much more informative if it showed still images of various phases of the two types of jump, so the reader can get a better understanding of what was done. This could be photographs of the participants or images of the motion capture data at these time points. You could even highlight the unweighted and propulsive phases here. Showing an image of the shoes is not particularly interesting, and one image of the motion capture markers with no context is also not very informative.

Response: We thank the reviewer for this comment. We added Figure 2 to represent the stick images and phase definitions as follows:

Figure 2. Definition of the unweighting and propulsive phases of single- and double-leg CMJs. Typical data and stick images are presented: vertical component of GRF (a), vertical CoM velocity, vertical CoM displacement (c), and stick images in sagittal and frontal planes (d, e). The circle markers (a-c) indicate when the CoM velocity was zero, representing the transition from the unweighting to the propulsive phase.

- Figure 2- Graphs showing the amount of countermovement performed in each jump and jump heights would be informative here, alongside vGRFs.

Response: We have added jump height of CMJs in Figure 3. The vertical CoM displacement was shown in Figure 8 to discuss the countermovement differences between single- and double-leg CMJs.

Figure 3. Mean and individual values of jump heights (a) and vertical component of GRF (b) during single-leg (blue solid line) and double-leg (red dashed line) CMJs. The circle markers indicate when the CoM velocity was zero, representing the transition from the unweighting to the propulsive phase.

Figure 8. The vertical CoM displacement (a), the maximum depth of the CoM displacement (b), and typical stick images at the maximum depth (c) during single-leg (blue) and double-leg (red) CMJs. The circle markers indicate when the CoM velocity was zero, representing the transition from the unweighting to the propulsive phase.

Reviewer #2

General comments: This study addresses an interesting and relevant question regarding the biomechanics of single- and double-leg countermovement jumps (CMJs), with potential implications for understanding athletic performance. However, the manuscript would benefit from careful revision of the English throughout, particularly with respect to grammar, spelling and sentence clarity. While the scientific content is largely clear, there are numerous minor language issues that reduce readability and occasionally obscure meaning, I strongly recommend a thorough language revision prior to resubmission.

Response: Thank you very much for your valuable comments and suggestions. We have addressed each point in our responses and revised the manuscript accordingly. In addition, we have carefully checked and revised the grammar, spelling, and overall clarity throughout the manuscript.

Abstract:

* The opening sentences would benefit from more clearly stating why this study is valuable. At present, the motivation for the work is not sufficiently established. Consider briefly emphasising the importance of SMJ performance in sport and why understanding its biomechanical determinants is meaningful.

Response: Thank you very much for this valuable suggestion. We revised the first and second sentences to clearly highlight the importance of single-leg jumping performance in sports and the value of understanding its biomechanical determinants.

Revised text (Page 3, Lines 49-53):

“Single-leg jumping tasks are directly linked to scoring in various sports; therefore, identifying the biomechanical determinants of higher single-leg jumping performance is crucial for athletic performance. Single-leg jumping can utilize the hip and lumbosacral joints in three dimensions. However, the joint kinetic variables in the frontal and horizontal planes associated with jump height are not fully understood.”

* While the main findings are described, the broader implications are not clearly articulated. It would strengthen the abstract to explicitly state why these results matter for biomechanics research or applied training practice.

Response: We appreciate this suggestion. We revised the latter half of the abstract to more clearly highlight the broader implications of our findings for biomechanics research and applied training practice.

Revised text (Page 3, Lines 61-64):

“The results highlight different mechanisms for achieving higher jump height between single- and double-leg CMJs. The findings suggest that strength training and movement modification targeting the frontal- and horizontal-plane motions may improve single-leg CMJ performance.”

Introduction: The introduction is generally well structured and addresses three key points:

1. Increasing understanding of the biomechanics of single- and double-leg CMJs is valuable. The importance and relevance of CMJs are clearly described in the first paragraph.
2. There are biomechanical differences between single- and double-leg CMJs. This is addressed in the second and third paragraphs, although this key message only becomes apparent several lines in. I recommend revising the opening sentences of these paragraphs to make this distinction clearer from the outset.
3. There is limited understanding regarding the timing of kinetic variables associated with jump height and identifying these may be important. This knowledge gap is outlined in paragraph four.

However, clarity could be improved in several places. Several sentences would benefit from rephrasing for clarity, particularly where technical concepts are introduced, and there are a number of minor grammatical issues (e.g. subject-verb agreement, repeated words) that should be addressed throughout the introduction.

Response: We appreciate the valuable suggestions and comments. We mainly revised the first and second paragraphs to emphasize the importance of biomechanical research on CMJs. In addition, we revised the Introduction throughout to improve clarity and corrected minor grammatical issues.

Specific points include:

* Line 68: clarify the phrasing (e.g. "countermovement jumps are..." or "jumping is...").

Response: We appreciate this suggestion. However, because the 1st and 2nd paragraphs were substantially reorganized during revision, we did not revise it.

* Lines 70-71: consider explicitly stating that improved biomechanical understanding may inform coaching and technique development, which may in turn enhance athletic performance.

Response: Thank you for this comment. To emphasize the practical importance of biomechanical research on CMJs, we added further explanation to the first paragraph.

Revised text (Page 4, Lines 67-71):

“Single-leg jumping tasks are frequently observed in a variety of sports activities, such as basketball, volleyball, soccer, and track and field. Single-leg jumping performance is directly linked to scoring and competitive outcomes (e.g., lay-up shots in basketball and jump distance in

the long jump). Thus, clarifying the biomechanical determinants of single-leg jumping performance may contribute to improving athletic performance.”

*** Line 75: this may be clearer if rephrased as "However, current understanding of ... is limited to...".**

Response: We appreciate this suggestion. However, because the paragraph was substantially reorganized during revision, we did not use this phrasing.

*** Lines 76-78: I found the explanation here to be a little unclear; please clarify.**

Response: We appreciate this comment. We revised the first and second paragraphs to clarify this point.

*** Lines 79-80: please clarify what is meant by "important determinants" (e.g. determinants of jump height?).**

Response: We appreciate this suggestion. However, because the paragraph was substantially reorganized during revision, we did not use this phrasing.

*** Line 80: "are" should be used instead of "is".**

Response: We appreciate this comment. We revised it (Page 4, Line 78).

*** Lines 87-88: consider revising to "...and powers have been observed...".**

Response: We appreciate this comment. We revised it (Page 4, Line 87).

*** Lines 94-95: the word "contribute" is used twice; revising this sentence may improve clarity and flow.**

Response: We appreciate this comment. We revised it.

Revised text (Page 4, Lines 94-96):

"... the free-leg side elevation of the pelvis contributes to the generation of mechanical energy related to jump height in single-leg squat jump, whereas it does not in double-leg squat jump"

Methods:

*** The participant cohort includes athletes who routinely perform jumping movements (e.g. volleyball players) and those who do not (e.g. undefined 'marine sports'). Differences in training background could plausibly influence technique and performance. Have analyses been conducted excluding participants who do not regularly train jumping movements and if so, are the results consistent? This issue is briefly acknowledged later in the manuscript, but I feel that it warrants stronger consideration here.**

Response: Thank you very much for this valuable suggestion. We conducted an additional analysis excluding the three marine-sport athletes ($n = 45$). The results were consistent with the main findings obtained from the full sample ($n = 48$). We have also added further discussion of this issue in the Limitations section. Although these additional results are not included in the revised manuscript, the corresponding figures are provided in this Author Response File.

Revised text (Page 14, Lines 398-403):

"Additionally, we recruited participants from several sports backgrounds. In particular, three marine-sport athletes (3/48 participants) were included; however, an additional analysis excluding these three athletes ($n = 45$) showed similar associations between jump height and joint kinetics to those observed in the full sample ($n = 48$). Because single-leg CMJs require three-dimensional motion of the hip and lumbosacral joints, the fundamental movement pattern may be similar across sex and sports backgrounds."

The results with $n=45$ samples (without three marine-sport athletes) as follows:

Joint angular velocities, torques, and powers of the ankle (a, f, and k), knee (b, g, and l), and hip (c-e, h-j, and m-o) joints during single-leg (blue solid line) and double-leg (red dashed line) CMJs. The circle markers indicate when the CoM velocity was zero, representing the transition from the unweighting to the propulsive phase. The asterisks (*) and horizontal bars at the top of each graph indicate the time points showing significant correlations with jump height. The asterisks and bars represent positive correlations, whereas the bars alone represent negative correlations. The blue and red lines represent single- and double-leg CMJs, respectively.

Joint angular velocities, torques, and powers of the lumbosacral joint during single-leg (blue solid line) and double-leg (red dashed line) CMJs. The circle markers indicate when the CoM velocity was zero, representing the transition from the unweighting to the propulsive phase. The asterisks (*) and horizontal bars at the top of each graph indicate the time points showing significant correlations with the jump height. The asterisks and bars represent positive correlations, whereas the bars alone represent negative correlations. The blue and red lines represent single- and double-leg CMJs, respectively.

*** No female participants were recruited. Given the well-documented sex bias in biomechanics and sports science research, this is a notable limitation. Please provide a justification for this decision and explicitly acknowledge it as a limitation.**

Response: We thank the reviewer for this comment. We have added this issue as a limitation in the Limitations section.

Revised text (Page 14, Lines 428-433):

“Second, although this study recruited male athletes from various sports backgrounds to enhance generalizability, sex and sports background may influence our findings. Previous studies have reported that GRF variables and joint kinematics differ between female and male athletes during jumping tasks (Cronström et al., 2016). Such sex-related differences may lead to differences in the joint kinetic determinants of CMJ jump height.”

*** The standardized footwear is shown in Figure 1a, but its characteristics are not described. Please clarify whether this was a specific model and provide relevant details (e.g. cushioning, sole stiffness).**

Response: We have added additional information about the footwear characteristics.

Revised text (Page 6, Lines 137-139):

“This footwear consisted of standard athletic gym shoes that did not have a rocker sole, excessive cushioning, and a carbon plate as recently seen in “super shoes (Healey et al., 2022; Hébert-Losier & Pamment, 2023)”.”

*** Several methodological procedures would benefit from clearer referencing to established protocols:**

o Warm-up procedure: does this follow a standard protocol? If so, please cite a reference.

Response: We thank the reviewer for this comment. The warm-up protocol was developed by the authors based on pilot testing conducted prior to data collection for this study. We have added this explanation in the Method section.

Revised text (Page 6, Lines 143-145):

“This warm-up protocol was designed by the authors and piloted prior to the experiment. The warm-up familiarized participants with the CMJ task and helped participants perform maximal-effort CMJs throughout experiments.”

o Line 165: please provide a reference to support the GRF threshold of <20 N.

Response: We have added the reference (Page 7, Line 169).

o Lines 178-179: “The whole body was modeled as a rigid-linked skeletal model...” Please provide a reference for this modelling approach. Also note the incorrect spelling of “modelled”.

Response: We have added the reference (Page 7, Lines 190-191).

o Lines 188-192: does this analysis follow an established protocol? If so, please cite it.

Response: We have revised this part to clarify the procedure and added the relevant reference (Page 8, Lines 201-202).

*** The countermovement depth was self-selected. Please discuss whether this lack of standardization may have influenced inter-individual variability and the observed correlations.**

Response: We appreciate this comment. We have discussed the inter-individual differences in countermovement and included an additional analysis (Figure 8).

Revised text (Page 13, Lines 383-388):

“Additionally, because CoM depth was not standardized between participants in this study, CoM displacement showed large inter-individual differences in both single- and double-leg CMJs: the ranges of maximum depth were -0.377 to -0.166 and -0.471 to -0.212, respectively. These inter-individual differences may be associated with inter-individual variability in joint kinetics during both unweighting and propulsive phases.”

*** For single-leg CMJs, the leg with the higher mean jump height was selected for analysis. Please justify this choice and clarify whether left-right comparisons were performed or considered.**

Response: To address this point, we additionally analyzed the single-leg CMJ on the inferior leg side (see Supplementary File 2: Table S1 and Figures S5-S8). We also added a discussion regarding the differences in the results between the superior and inferior leg sides.

Revised text (Pages 13-14, Lines 400-413):

“Furthermore, although the present study analyzed the leg with the higher jump height (the superior leg) in the single-leg CMJ to represent each participant’s best performance, an additional analysis of the contralateral leg (the inferior leg) showed both similarities and differences in the correlation results between the superior and inferior legs (Supplementary File 2: Table S1 and Figures S5-S8). The inferior leg did not show a significant time interval in which hip abduction torque and abduction-adduction power were correlated with jump height. In contrast, the peak values of hip abduction torque, hip adduction-abduction power, and lumbosacral lateral flexion torque were positively associated with the inferior-leg jump height, consistent with those of the superior leg. Therefore, these findings indicate that greater hip abduction and lumbosacral lateral flexion torques and their powers were associated with higher jump height of the single-leg CMJ, even in the inferior leg. However, inter-limb differences in countermovement characteristics or postural stability may influence the kinetic determinants of jump height; thus, inter-limb differences in jumping mechanics should be examined in future studies.”

Statistical analysis: Correlation analyses may be appropriate for the stated aims, but please consider whether additional or complementary analyses (e.g. accounting for multicollinearity or shared variance between joints) could provide further insight.

Response: We appreciate this suggestion. We added the results of multiple regression analyses to clarify the differences in the kinetic predictors between single- and double-leg CMJs (Tables 2 and 3). We also revised the Methods and Discussion sections accordingly. We believe that these additional results strengthen our findings.

Revised text in Methods (Page 8, Lines 222-230):

“Moreover, a stepwise multiple regression analysis was performed for each CMJ task to examine whether the predictive models for jump height differed between single- and double-leg CMJs. In this analysis, the jump height was set as the dependent variable, and the peak torques and powers of lower-limb joints and lumbosacral joint were set as independent variables. To minimize the effects of multicollinearity among predictors, especially between torque and power, the stepwise multiple regression analyses were performed separately using peak torques and peak powers. The correlation and stepwise multiple regression analyses were performed using the Statistics and Machine Learning Toolbox in MATLAB 2020b.”

Revised text in Results (Page 11, Lines 302-307):

“A multiple regression analysis using peak torques (Table 2) showed that the predictors of jump height differed between the single- and double-leg CMJs. In the double-leg CMJ, the predictors were peak ankle plantarflexion and hip extension torques, whereas in the single-leg CMJ, the predictors were peak ankle plantarflexion, hip abduction, lumbosacral extension, and lateral flexion torques (Table 2). Additionally, the same predictors were selected in the analysis using peak powers in both CMJs (Table 3).”

Revised text in Discussion (Page 12, Lines 343-351):

“Moreover, the multiple regression analysis using peak joint powers selected flexion-extension powers as the predictors in the single- and double-leg CMJs. In contrast, the multiple regression analysis using peak torques showed that the jump height of the single-leg CMJ was predicted by ankle plantarflexion, hip abduction, lumbosacral extension, and lumbosacral lateral flexion

torques, but the hip and knee extension torques were not selected. These predictors differed from those for the double-leg CMJ as follows: jump height was predicted by ankle plantarflexion and hip extension torques. These results suggest that the hip and knee extension torques have a weaker association with jump height in the single-leg CMJ than in the double-leg CMJ.”

New Tables:

Table 2. Results of stepwise multiple regression analyses using peak torques of lower-limb joints and lumbosacral joint to predict jump height in the single- and double-leg CMJs.

Independent variables	β	Equation	Adjusted R^2
Single-leg CMJ			
x_1 : ankle plantarflexion	0.510 ($p < 0.001$)	y	0.669
x_2 : hip abduction	0.267 ($p = 0.017$)	$= 0.100x_1$	(p < 0.001)
x_3 : lumbosacral extension	0.233 ($p = 0.011$)	$+ 0.046x_2$	
x_4 : lumbosacral lateral flexion	0.186 ($p = 0.038$)	$+ 0.021x_3$	
		$+ 0.038x_4$ $- 0.169$	
Double-leg CMJ			
x_1 : ankle plantarflexion	0.498 ($p < 0.001$)	y	0.432
x_2 : hip extension	0.347 ($p = 0.004$)	$= 0.187x_1$	(p < 0.001)
		$+ 0.056x_2$ $- 0.005$	

Note. β , standardized partial regression coefficient; y , jump height.

Table 3. Results of stepwise multiple regression analyses using peak powers of lower-limb joints and lumbosacral joint to predict jump height in the single- and double-leg CMJs.

Independent variables	β	Equation	Adjusted R^2
Single-leg CMJ			
x_1 : ankle plantar-dorsi flexion	0.546 ($p < 0.001$)	y	0.779
x_2 : knee flexion-extension	0.270 ($p = 0.003$)	$= 0.007x_1$	(p < 0.001)
x_3 : hip flexion-extension	0.247 ($p = 0.002$)	$+ 0.005x_2$	
x_4 : lumbosacral flexion-extension	0.195 ($p = 0.009$)	$+ 0.004x_3$	
		$+ 0.006x_4$ $- 0.028$	
Double-leg CMJ			
x_1 : ankle plantar-dorsi flexion	0.497 ($p < 0.001$)	y	0.792
x_2 : lumbosacral flexion-extension	0.383 ($p < 0.001$)	$= 0.012x_1$	(p < 0.001)
x_3 : knee flexion-extension	0.355 ($p < 0.001$)	$+ 0.010x_2$	
x_4 : hip flexion-extension	0.175 ($p = 0.018$)	$+ 0.009x_3$	
		$+ 0.006x_4$ $- 0.038$	

Note. β , standardized partial regression coefficient; y , jump height.

Results:

* The Results section is quite dense and may be difficult for readers to follow. Consider restructuring this section using clearer subheadings and/or shorter paragraphs to guide the reader.

Response: We have revised the entire Results section to present the findings more clearly.

* Figure 2 contains a spelling error ("Sigle-leg CMJ").

Response: We appreciate this comment. We corrected the spelling error in Figure 3.

* Please ensure that all figures and tables are clearly referenced at appropriate points in the main text.

Response: We appreciate this comment. We checked and corrected the figure and table citations throughout the manuscript.

Discussion:

* The interpretation of frontal- and horizontal-plane kinetics in single-leg CMJs is thoughtful and well linked to prior literature. However, some statements regarding applicability to sports performance (e.g. running, high jump, long jump) should be more cautiously framed, given that not all participants were trained jump specialists. This limitation is acknowledged briefly, but I recommend expanding this discussion to more clearly delimit the populations to which the findings can be generalized.

Response: We have expanded the discussion of the study limitations to more carefully delimit the populations to which our findings can be generalized, including the lack of female participants and the inclusion of marine-sport athletes.

* The Discussion occasionally reiterates the same findings across multiple paragraphs. Condensing these sections and focusing more explicitly on interpretation and implications would improve clarity and flow.

Response: Thank you for this valuable suggestion. We revised the Discussion section throughout to improve clarity and flow by reducing repetition.

* While the authors note the cross-sectional design, several parts of the Discussion describe correlational relationships in mechanistic or causal terms. I recommend more consistently framing these findings as associations and clarifying that the observed kinetic patterns may reflect successful strategies rather than direct drivers of performance.

Response: We appreciate this suggestion. We revised the Discussion section throughout to more consistently describe these findings as associations.

* Statements extending the findings to improvements in other single-leg motor tasks (e.g. running or sport-specific performance) should be more clearly framed as speculative, as these outcomes were not directly assessed in the present study.

Response: We revised the relevant statement to more clearly present this point as speculative.

Revised text (Page 14, Lines 422-424):

“Therefore, in addition to strength training, enhancing the neuromuscular coordination in the hip and lumbosacral joints may be associated with higher single-leg CMJ performance, and may also have implications for performance in other single-leg motor tasks.”

In summary, this is a promising and technically strong study that addresses an important gap in the literature. Addressing the points above would substantially improve clarity, methodological transparency and interpretability.

Response: Thank you very much for your positive and valuable comments and suggestions. We believe that our additional analysis and discussion enhance our findings.

Second decision letter

MS ID#: bio.062434R1

MS Title: Differences in three-dimensional kinetic determinants of jump height between single- and double-leg countermovement jumps

Authors: Terumitsu Miyazaki; Shota Yamamoto; Hirotomo Kubota

This morning I have read through your responses to the reviewer comments and the associated edits to the manuscript. I am happy to tell you that your manuscript has been accepted for publication in Biology Open, pending our standard publication integrity checks. It was accepted on 9th April 2026.